# Carbapenem-Resistant *Enterobacteriaceae* Bacteremia in Pediatric Patients in Latin America and the Caribbean: A Systematic Review and Meta-Analysis

**DOI:** 10.3390/antibiotics13121117

**Published:** 2024-11-22

**Authors:** Silvina Ruvinsky, Carla Voto, Macarena Roel, Victoria Portillo, Gabriela Naranjo Zuñiga, Rolando Ulloa-Gutierrez, Daniel Comandé, Agustín Ciapponi, Gabriela Aboud, Martín Brizuela, Ariel Bardach

**Affiliations:** 1Coordinación de Investigación, Hospital de Pediatría Dr. Juan P. Garrahan, Buenos Aires 1245, Argentina; 2Instituto de Efectividad Clínica y Sanitaria, Buenos Aires 1414, Argentinaaciapponi@iecs.org.ar (A.C.);; 3Hospital Nacional de Niños Dr. Carlos Sáenz Herrera, San José 10103, Costa Rica; 4Centro de Investigaciones Epidemiológicas y Salud Pública (CIESP-IECS) CONICET, Buenos Aires 1414, Argentina; 5Unidad de Pediatría, Hospital General de Agudos Vélez Sarsfield, Buenos Aires 1550, Argentina; martin.brizuela1984@gmail.com

**Keywords:** carbapenem-resistant *Enterobacteriaceae*, bacteremia, children, Latin America and the Caribbean, meta-analysis, systematic review

## Abstract

**Background:** Data on the health impact of carbapenem-resistant *Enterobacteriaceae* bloodstream infections (CRE-BSIs) in pediatric populations from Latin America and the Caribbean (LAC) are limited. This systematic review aims to examine the demographic, clinical, and microbiological aspects and resource utilization of this infection in children from this region. **Methods:** This systematic review investigates the impact of CRE-BSIs in pediatric populations across LAC. Following the Cochrane methodology and PRISMA/MOOSE guidelines, we conducted an extensive search of different databases, including MEDLINE/PubMed, LILACS (SciELO), CENTRAL, CINAHL, Embase (Ovid), the Cochrane Library, and the World Health Organization (WHO) database, and relevant websites for published articles between January 2012 and September 2024. The review included studies on hospitalized patients under 19 years of age with CRE-BSIs. **Results:** Fourteen studies involving 189 patients were analyzed. Most cases were reported from Brazil, Argentina, Colombia, and Paraguay. The median age of the patients was 35 months. Key risk factors included immunocompromised status, invasive procedures, carbapenem use, and colonization. The infections were predominantly hospital-acquired, with *Klebsiella pneumoniae* and *Serratia* spp. being the most common pathogens. KPC and NDM were the primary resistance mechanisms. Most patients received combination antimicrobial therapy for a median of 17 days. An alarmingly high mortality rate at 34% was found. **Conclusions:** Our findings highlight that CRE-BSIs pose a significant threat to children with underlying conditions in LAC, leading to substantial morbidity and mortality. Implementing robust antimicrobial stewardship programs and effective infection control measures are crucial to curbing the spread of CRE-BSIs in the region. This review underscores the need for targeted interventions and further research to address this critical public health concern in pediatric populations across LAC.

## 1. Introduction

Antimicrobial resistance (AMR) poses a critical challenge in countries from Latin America and the Caribbean (LAC), contributing significantly to global mortality, with 4.95 million deaths in 2019. This underscores the urgent need for effective control strategies [1]. In LAC, infections caused by multidrug-resistant organisms (MDROs) have a case fatality rate (CFR) of 45%, which is even higher among patients receiving inappropriate empirical antibiotic treatment [2].

The incidence of carbapenem-resistant *Enterobacteriaceae* (CRE) infections in children has been increasing globally [3], with CRE bloodstream infections (CRE-BSIs) predominantly occurring in healthcare settings [4]. CRE-BSIs have become particularly prevalent in pediatric intensive care units (PICUs), significantly affecting morbidity, mortality, and public health systems [5].

The COVID-19 pandemic has exacerbated the situation, with increased carbapenem resistance likely due to inappropriate broad-spectrum antibiotic use [6,7]. Several countries reported the emergence of previously unidentified carbapenemase-producing *Enterobacteriaceae* and isolates co-expressing two or more enzymes including NDM, KPC, and OXA-48, or combinations of these [8]. The mechanism for AMR in Gram-negative bacilli against carbapenems involves the cross-transmission of resistance genes, primarily through conjugation—the most common of which with the genetic material, usually plasmids, being directly transferred between two bacteria via a sex pilus—transformation, and transduction. The horizontal transmission of carbapenem-resistance genes presents a serious problem in clinical settings, as it can lead to the rapid dissemination of resistance among different *Enterobacteriaceae* species, limiting the available therapeutic options [9].

CRE has spread significantly in LAC countries, with some regions becoming endemic for certain carbapenemases. KPC is the predominant carbapenemase gene in Latin America, being endemic in Brazil, Colombia, Argentina, and Puerto Rico. NDM has been reported in Brazil, Colombia, Mexico, and Guatemala, while OXA-48 has been detected in Venezuela, Brazil, and Argentina [10]. Carbapenemase-producing *Enterobacteriaceae* (CPE) exhibit resistance not only to carbapenems but also to multiple antibiotic classes, including all types of β-lactams and β-lactamase inhibitor combinations, fluoroquinolones, aminoglycosides, and tetracyclines. This extensive multi-drug resistance severely limits therapeutic options for CPE infections [11].

CRE-BSI is associated with prior PICU admission and extended hospital stays, affecting resource utilization and healthcare costs [12]. The 28-day mortality rate for CRE-BSI is significantly higher than that for infections caused by sensitive *Enterobacteriaceae* [13].

Understanding the impact of CRE-BSIs on disease burden is crucial for informed public health decision-making. Key strategies include strengthening epidemiological surveillance, implementing antimicrobial stewardship programs, and promoting responsible antibiotic use [1].

While most CRE-BSI research focuses on adults, data on pediatric populations in LAC remain limited. Our systematic review and meta-analysis aimed to comprehensively examine the epidemiological data, risk factors, clinical manifestations, microbiological profiles, resistance patterns, and resource utilization associated with CRE-BSIs in children from LAC, incorporating both national and regional epidemiological reports.

## 2. Results

The literature search strategy yielded 928 potential studies, of which 14 met the eligibility criteria (Figure 1) and were included in the final analysis (Table 1). The search strategy is presented in the Supplementary Annex, PRISMA 2020 checklist, and the list of excluded studies together with the reason for exclusion in Appendix A.

Our analysis encompassed 14 studies, totaling 189 cases (Table 2). The research was conducted across several Latin American countries: Brazil (4 studies), Argentina (2), Colombia (3), Paraguay (2), Peru (1), Mexico (1), and Ecuador (1). The studies comprised nine case series, four case reports, and one case–control study.

The patients had a mean age of 35 months (95% CI: 16–76 months), with 56% being male (95% CI: 46–66%). At least one comorbidity was observed in all the patients, the most frequent of which were onco-hematological diseases (31%, 95% CI: 7–61%), liver transplantation (26%, 95% CI: 14–41%), and prematurity (22%, 95% CI: 2–82%). Notably, 62% of the patients were immunocompromised (95% CI: 36–95%).

Key risk factors for developing CRE-BSIs included invasive procedures (91%, 95% CI: 63–100%), hospitalization (86%, 95% CI: 52–99%), broad-spectrum antibiotic use (91%, 95% CI: 76–100%), and PICU admission (42%, 95% CI: 3–87%). Common invasive procedures were central venous catheter placement (100%, 95% CI: 92–100%), bladder catheterization (66%, 95% CI: 46–84%), surgery (59%, 95% CI: 35–80%), and mechanical ventilation (51%, 95% CI: 31–70%).

The majority of patients demonstrated recent antibiotic exposure prior to infection. Specifically, 84% (95% CI: 74–93%) had been exposed to carbapenems and 78% (95% CI: 15–100%) to third-generation cephalosporins within the three months preceding infection. Additionally, among patients who underwent colonization screening, 41% (95% CI: 2–87%) were found to have prior colonization with CRE.

The vast majority of CRE-BSI cases (97%, 95% CI: 77–98%) were healthcare-associated, with an average pre-infection hospital stay of 35 days. Most infections (64%, 95% CI: 25–96%) were secondary to an intra-abdominal (75%, 95% CI: 27–100%) or cutaneous (32%, 95% CI: 0–80%) focus. Primary bacteremias accounted for 26% of cases, and sepsis occurred in 81% of patients.

*Klebsiella pneumoniae* was the predominant pathogen (95%, 95% CI: 78–99%), followed by *Serratia* spp. (67%, 95% CI: 1–100%), *Escherichia coli* (9%, 95% CI: 1–51%), *Enterobacter cloacae* (2%, 95% CI: 0–12%), and *Citrobacter* spp. (2%, 95% CI: 0–12%). The main resistance mechanisms were KPC (97%), NDM (32%), and OXA (29%).

Regarding the use of resources, most patients received a combination of antimicrobial therapy (86%, 95% CI: 44–100%). The median length of the hospital stay was 30 days (IQR: 13–66 days), and the median length of the PICU stay was 16 days (IQR: 15–17 days). The median duration of antibiotic treatment was 17 days (IQR: 12–24 days). Overall, 78% (95% CI: 25–100%) of the patients required mechanical ventilation while 48% (95% CI: 7–91%) received vasoactive drugs. The mortality rate was 34% (95% CI, 19–49%).

### 2.1. Analysis of Epidemiological Reports

Epidemiological surveillance reports from the following countries were included: Argentina (4), Brazil (4), Colombia (3), Paraguay (2), Nicaragua (2), Guatemala (1), Ecuador (1), Mexico (1), Peru (1), and Venezuela (1). These reports included 504 microbiological isolates from 2012 to 2021 (Table 3).

### 2.2. Risk of Bias Assessment

Among the nine case series, eight were evaluated as having a low RoB. The domain that presented the most significant challenge in evaluation was determining whether the included cases were consecutive (Appendix A). The only case–control study was rated with a low RoB. The problematic domains were sample size calculation, randomization, and the blinding of observations, all related to the study design (Appendix A).

## 3. Discussion

We conducted this systematic review to characterize the CRE-BSIs in the pediatric population from LAC. Brazil, Argentina, and Colombia were the countries most represented in the region. The majority of studies were case series with substantial heterogeneity.

National reports on CRE-BSIs in LAC countries are often incomplete due to the limited availability of data. This is likely a result of the absence of mandatory reporting and surveillance systems for CRE in many regions. The lack of standardized monitoring means that important data, especially on pediatric CRE-BSI cases, are frequently missing or underreported. This contributes to the overall scarcity of information and highlights the need for improved reporting practices across the region.

In agreement with our findings, most pediatric reports have shown a median age under 5 years [28,29]. Other series, which included mostly patients who had cancer or those undergoing hematopoietic stem cell transplantation (HSCT), found an older median age [27].

Most children had at least one underlying condition, including hematologic malignancies, congenital anomalies, and prematurity [29,30]. Van Loon et al. found that comorbidities increased the risk of CRE-BSIs by almost three times [31].

Our analysis revealed a significant association between CRE-BSIs and prior invasive procedures, including central venous catheter placement, urinary catheter placement, and mechanical ventilation. A notable study by Pannaraj et al. found that an overwhelming majority—approximately 90%—of patients with CRE-BSIs had indwelling medical devices [32].

A crucial finding was that the vast majority of the CRE-BSI cases, ranging from 75% to 92%, had a history of exposure to broad-spectrum antibiotics. Carbapenems and third-generation cephalosporins were the most frequently implicated antibiotic classes. This underscores the potential role of antibiotic pressure in selecting for resistant organisms [22,23]. While our focus is on the pediatric population, it is worth noting that similar trends have been observed in adults. Studies in adult populations have shown that previous exposure to beta-lactam antibiotics or carbapenems increases the risk of CRE infection by two-fold and five-fold, respectively [31]. These findings highlight the critical importance of antibiotic stewardship programs (ASPs), particularly regarding broad-spectrum agents, in preventing the emergence and spread of CRE infections. There is a need for the careful consideration of antibiotic use in high-risk groups, including immunocompromised patients and those with complex medical conditions. Ruvinsky et al. demonstrated that ASP implementation in a tertiary pediatric hospital from Argentina successfully reduced inappropriate antibiotic prescriptions, and these findings underscore the effectiveness of ASPs in improving antibiotic prescription practices for hospitalized children, with significant implications for enhancing patient care, reducing antimicrobial resistance, and optimizing resource utilization in pediatric healthcare settings [33].

We found that half of the infected patients were previously colonized. Akturk et al. showed that one-third of the previously colonized patients had a subsequent infection [34]. Among these patients, metabolic disorders, surgery, and neutropenia were associated with infection [35]. Previous intestinal colonization is a well-recognized predictor for bacteremia among immunocompromised children; therefore, the implementation of active surveillance and a reduction in the risk of colonization are necessary [36].

The majority of patients who developed CRE-BSIs were in hospital settings, as the infection is prevalent among PICU patients. Multiple studies, such as those conducted by Zhang et al. and Ara-Montojo et al. [28,29], have reported that nearly two-thirds of CRE-BSIs are acquired in hospital settings, while one-third is healthcare-related. A hospital stay of over 28 days was necessary before CRE-BSIs was linked to this type of infection [27,28].

Our analysis revealed that intra-abdominal infections were the predominant source of bacteremia in CRE-BSIs; however, other significant sources, such as pneumonia, catheter-associated infections, and infections of the urinary tract, have been described in the literature [37,38].

*Klebsiella pneumoniae* emerged as the most frequently isolated CRE, followed by *Enterobacter* and *Escherichia coli*, as reported by Liu et al. [30]. The most commonly observed resistance mechanism was KPC, followed by NDM and OXA. Nevertheless, the geographical distribution of the resistance mechanisms of CRE-BSIs in LAC is heterogeneous; certain regions have been more impacted by specific carbapenemase-producing pathogens, while, in others, they remain undetected [39]. In addition, there have been sporadic reports of *Enterobacteriaceae* isolates that produce two or more carbapenemases [40,41]. Another aspect to consider are the increasing interactions between humans and domestic animals, as documented cases of the transmission of carbapenemase-producing pathogens among these animals suggest a potential for the spread from animals to humans [42].

An additional relevant aspect of CRE infections is the high cost to the public and private healthcare system. Studies comparing CRE vs. carbapenem-sensitive *Enterobacteriaceae* (CSE) infections in the adult population found that patients with CRE infections are associated with a significantly longer length of hospital and ICU stay, although the proportion of patients that required admission was similar [43].

The failure to clear BSIs and a minimal inhibitory concentration (MIC) higher than eight mg/L for carbapenem were associated with mortality. On the other hand, the use of two or more effective drugs in combination reduced the mortality rate [34,37]. Mortality rates ranged from 7 to 17% on day 7 and 19 to 52% on day 28. Specifically, mortality at day 28 was associated with the requirement of mechanical ventilation and septic shock [27,44], as well as an inappropriate empirical antibiotic treatment and a delay in the adjustment of treatment [28]. CRE-BSIs were related to a significant increase in days of fever, PICU admission, and death in children with cancer or undergoing HSCT compared with CSE-BSIs [29].

We found a mortality rate of 34%; similar rates (18–52%) were reported in recent studies [28,37].

### 3.1. Limitations

Acknowledging certain limitations is essential for interpreting our findings. Firstly, there is a need for more studies and reports on CRE-BSIs in the pediatric population in LAC. The included studies often have a limited number of patients, and the clinical heterogeneity among these studies may affect the generalizability of our results. Furthermore, our reliance on retrospective data presents challenges, including incomplete records and variations in reporting practices. Moreover, our study did not identify specific data on the prevalence or incidence of CRE-BSIs.

As is common in meta-analyses, we encountered significant levels of heterogeneity, particularly in observational studies, where both the frequency and severity of heterogeneity tend to be more pronounced compared to experimental studies conducted under controlled conditions. This diversity arises from the complexity of real-world epidemiological research, encompassing varied populations, environmental exposures, and healthcare contexts. In such cases, traditional strategies to explore heterogeneity may not fully resolve these differences. Therefore, we used a random-effects model, particularly useful for assessing the variability among studies, producing wider, more cautious confidence intervals that represent a conservative approach. Nevertheless, in instances of high heterogeneity, the central estimate may still be misleading, and the confidence intervals, which are weighted and more reliable, should be prioritized for interpretation.

### 3.2. Contributions of This Study

Regardless of these limitations, our study provides important insights into the epidemiological and clinical aspects of CRE-BSIs in the pediatric population from LAC. The inclusion of diverse countries enhances the generalizability of our findings and provides a comprehensive overview of the regional landscape. This study highlights the need for the standardized reporting of drug utilization and drug resistance surveillance, the integration of current guidelines into clinical practice, and the optimization of resource allocation for effective CRE-BSI management. Future research should address these gaps to improve outcomes and report evidence-based practices.

## 4. Material and Methods

This systematic review and meta-analysis rigorously adhered to the Cochrane methodology [45] and conformed to the 2020 Preferred Reporting Items for Systematic Reviews and Meta-Analyses (PRISMA) guidelines for result reporting [45]. The protocol was duly registered in the PROSPERO database at the University of York (CRD42022352504).

We conducted searches across multiple databases, including MEDLINE/PubMed, Embase (Ovid), LILACS (SciELO), CENTRAL, CINAHL, the Cochrane Library, and the World Health Organization (WHO) database, and relevant websites, for articles published between 1 January 2012 and 30 September 2024. We manually searched the reference lists of the included articles for additional information. To ensure data integrity, we prioritized publications with the largest sample sizes when multiple articles reported on the same population, consulting with the principal investigator when necessary. After three unsuccessful attempts to contact authors for additional or missing information, references were excluded. We also reviewed the proceedings of international, national, or regional (LAC) scientific meetings and epidemiological surveillance reports. An annotated search strategy for grey literature was included to retrieve relevant information. All studies were identified without language restrictions. This methodological approach ensures a rigorous and comprehensive examination of the literature on CRE-BSIs in pediatric populations in the LAC region, providing a strong foundation for our analysis and conclusions.

### 4.1. Eligibility Criteria

Both comparative and non-comparative studies were included, regardless of their publication status or language. Studies that provided insufficient data, narrative reports, and case series reviews, as well as articles lacking full-text access or classified as systematic reviews (SRs), were excluded. The reference lists of relevant SRs were reviewed to identify additional studies of interest. We focused on studies providing data on hospitalized patients under the age of 19 with CRE-BSIs (involving *Klebsiella*, *Enterobacter*, *Escherichia coli*, *Proteus*, and *Serratia species*).

### 4.2. Outcomes of Interest

We examined epidemiological outcomes such as prevalence, incidence, and mortality. We also looked at the age of the patients, any underlying conditions, prior history of invasive procedures, colonization, microbiological profile, antimicrobial-resistance patterns, antibiotic treatment, PICU admission, length of stay, and resource use. We calculated outcomes for all available patients.

### 4.3. Study Selection, Data Extraction, and Assessment of the Risk of Bias in Included Studies

The articles were screened by pairs of reviewers who independently evaluated study titles and abstracts. Any discrepancies were resolved through consensus among the whole team. We used COVIDENCE software (https://www.covidence.org/ accessed on 10 October 2024) for the initial phases of the systematic review [46,47,48]. Additionally, we examined the reports of passive surveillance report systems from the Pan American Health Organization (PAHO) and LAC countries.

We looked at all the potentially relevant studies, and two reviewers independently evaluated the risk of bias (RoB). Any disagreements were resolved through discussion among the team. We used an online spreadsheet to collect the data. We extracted information about the studies (publication type, year, authors, location, study design, and risk of bias assessment method), population characteristics, and outcomes (incidence rate, specific mortality, and fatality rate). We contacted the authors for additional information when needed.

The RoB in observational studies and in the control arms of trials was evaluated using a checklist designed by the United States National Heart, Lung, and Blood Institute (NHLBI), which categorizes studies into high (Poor), moderate (Fair), and low (Good) RoB [49]. For cohort and cross-sectional studies, the tool includes 14 items, while it consists of 9 items for case series studies and 12 items for case–control studies. Each criterion was classified as “low risk”, “high risk”, or “uncertain risk”, and was accompanied by a descriptive summary of the information that guided our assessment. If a criterion was rated as “uncertain”, we reached out to the study authors for clarification. Pairs of independent reviewers evaluated the RoB, and any discrepancies were resolved through consensus within the research team.

### 4.4. Data Synthesis

Proportion meta-analyses were conducted to analyze the data. An arc-sine transformation was applied to stabilize the variance of proportions using the Freeman–Tukey variant of the arc-sine square-root-transformed proportions method
$$y=arcsin\;\left(\sqrt{\frac{r}{n+1}}\right)+arcsin\;\left(\sqrt{\frac{r}{\left(n+1)(n+1\right)}}\right)$$
with the variance of
$$\frac{1}{n+1}$$
where n is the population size [46]. We calculated pooled proportions by back-transforming the weighted mean of the transformed proportions, using inverse arcsine variance weights for both the fixed and random-effects models. When we found heterogeneity between studies, we applied Der Simonian–Laird weights for the random-effects model. To measure the proportion of overall variation attributable to between-study heterogeneity, we calculated the I^2^ statistic. An I^2^ value greater than 60–70% indicated substantial heterogeneity, while a value below 30% indicated a low level of heterogeneity.

We used the R software (version 4.3) package meta and its functions meta mean, metaprop, and forest.meta. Data were synthesized using descriptive and meta-analytic approaches by outcome measures (RRs or Mantel–Haenszel ORs) or Peto ORs for dichotomous data. For continuous data, mean difference (MD) and 95% confidence interval (CI) were reported for all outcomes. In cases where it was impossible to calculate association measurements, we used simple descriptive statistics.

## 5. Conclusions

CRE-BSIs cause significant morbidity, mainly in pediatric patients with underlying diseases in the LAC region. CRE-BSIs have a high public health impact. Adequate policies and programs are required to optimize antimicrobial use and avoid multidrug resistance.

## Figures and Tables

**Figure 1 antibiotics-13-01117-f001:**
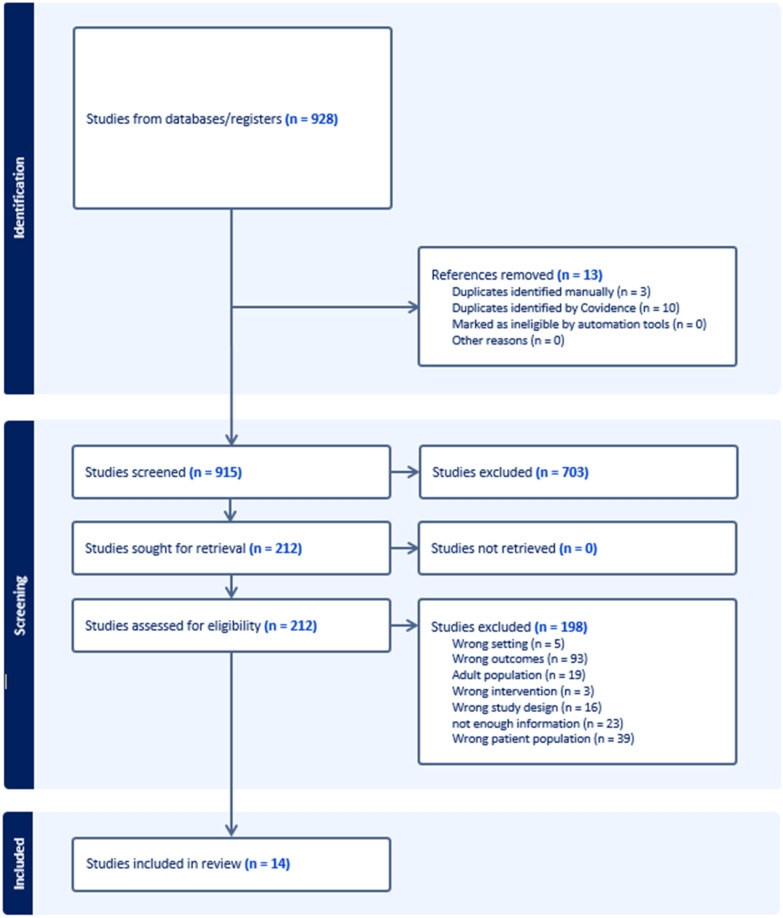
PRISMA 2020 study flow diagram.

**Table 1 antibiotics-13-01117-t001:** Characteristics of included studies (n = 14).

Author and Year of Publication	Country	Study Design	Number of Patients	Male n	Age in Months Median (IQR)	Underlying Condition n (%)	Colonization n	Mortality n	Period
Martino, 2019 [14]	Argentina	Case report	1	1	48	Erythodermic psoriasis	Yes (during episode)	1	February 2014
Ruvinsky, 2022 [15]	Argentina	Case–control	46	23	36 (11.2–117)	45: hematological diseases (23), liver transplantation (12)	35	6	January 2014–December 2019
Alvares, 2019 [16]	Brazil	Case series	6	5	19.2 (12–70)	Non-specified (5)	2	4	January 2012–December 2016
de Oliveira, 2015 [17]	Brazil	Case series	4	3	17 (5–28)	4: cancer (2)	NR	1	October 2009–June 2013
Higashino, 2018 [18]	Brazil	Case series	2	2	180	HSCT	NR	2	2008–2015
Nascimento, 2022 [19]	Brazil	Case report	1	1	1	Prematurity	NR	0	NR
Marquez Herrera, 2016 [20]	Colombia	Case series	10	NR	48	Hematological diseases (5)	No	4	2012–2015
Perez, 2020 [21]	Colombia	Case series	14	7	117 (32–177)	HSCT	5	4	January 2012–January 2017
Reyes Chacón, 2021 [22]	Ecuador	Case series	41	23	12 (11–24)	Congenital heart disease (18), prematurity (15), cancer (4)	NR	24	May 2016-July 2018
Aquino Andrade, 2018 [23]	Mexico	Case series	3	NR	72	Leukemia	NR	3	April 2016–July 2016
Rodriguez Palacios, 2016 [24]	Paraguay	Case report	1	1	NR	NR	NR	NR	April 2015–September 2015
Touchet, 2021 [25]	Paraguay	Case series	3	3	4.4	NR	NR	NR	February 2021–September 2021
Barrios, 2021 [26]	Perú	Case report	1	0	NR	No	NR	NR	January 2019–December 2019
López Cubillos, 2023 [27]	Colombia	Case series	56	36	126 (60–176)	HSCT	50	20	2012–2021

NR: not reported. HSCT: hematopoietic stem cell transplantation.

**Table 2 antibiotics-13-01117-t002:** Characteristics of included population (n = 189).

Variables	Pooled Estimates from Meta-Analysis % (95% CI)	I^2^ (%)
Risk factors associated with CRE-BSI		
- Underlying conditions	100 (92–100)	65
- Previous CRE colonization	41 (2–87)	92
- Broad spectrum antibiotics *	91 (76–100)	63
- Carbapenem use *	84 (74–93)	0
- Surgical procedures *	59 (35–80)	71
- Central venous catheter *	100 (92–100)	3
- Mechanical ventilation *	51 (31–70)	0
- Bladder catheter *	66 (46–84)	0
Characteristics of infection and resource use		
- Bacteremia secondary to intra-abdominal focus	75 (27–100)	64
- Combined antibiotic treatment	86 (44–100)	86
- Length of total hospital stay, days Median (IQR)	30 (13–66)	99
- Mortality rate	34 (19–49)	14

* up to 3 months before CRE-BSI.

**Table 3 antibiotics-13-01117-t003:** Bacterial distribution in 504 microbiological isolates from 20 epidemiological reports.

Microbiological Isolates	n (%)
*Klebsiella pneumoniae*	460 (91)
*Escherichia coli*	23 (4.5)
*Serratia marcescens*	8 (1.6)
*Enterobacter cloacae*	5 (1)
*Enterobacter* spp.	4 (0.8)
*Klebsiella oxytoca*	2 (0.4)
*Citrobacter* spp.	2 (0.4)

## Data Availability

Data are contained within the article and Appendix A.

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
