# Peer review of "Carbapenem-Resistant Enterobacteriaceae Bacteremia in Pediatric Patients in Latin America and the Caribbean: A Systematic Review and Meta-Analysis"

_antibiotics, 2024, doi:10.3390/antibiotics13121117_

Round 1
Reviewer 1 Report
Comments and Suggestions for Authors
Dear Authors,
I congratulate all the Authors for their contributions to the writing of the manuscript entitled “Carbapenem-Resistant Enterobacteriaceae Bacteremia in Pediatric Patients in Latin America and the Caribbean: A Systematic Review and Meta-Analysis”.
I have a few comments on the manuscript:
1. Abstract: Please list all the databases used to search related articles in line 20. Also, please confirm the year 2012 is correct? 2012 to 2022 (10 years) for the comprehensive search? Did the authors mean articles published between Jan 2012 and Sept 2022? If so, please rephrase the sentence.
2. Line 25: Organism names should be in italics. Please review the manuscript thoroughly.
3. Abstract: Results: Results are too general. Please highlight your important results from the meta-analysis in this section.
4. Introduction: Line 52: Perhaps a list of the isolated enzymes would benefit the readers.
5. Line 53: Please elaborate more on the mechanism of AMR spread via horizontal gene transfer.
6. Line 55: Which regions? This will benefit the readers when reading and comparing with your results later.
7. Line 56: Multiple antibiotic families such as?
8. Methods: Line 77: again, please verify the year that you performed the searches. 10 years is too long. Is this correct? Or are you referring to these dates as the dates for articles published? Otherwise, you may change it to “for articles published between January 1, 2012 and September 30, 2022.
9. Also, is there any specific reason why the recent articles up until September 2024 are not included in the analysis?
10. Figure 1 (PRISMA flow chart) should be moved to the Method section.
11. Line 128: Please write the formula in correct format and centered on the page, not in line with text.
12. Line 134: It should be I2, not I2.
13. Line 135: Use an en dash to denote range, not a hyphen. Please review the manuscript thoroughly.
14. Figure 1: Please list the number of studies identified in each database used.
15. Figure 1: studies excluded (n = 653), based on what criteria? Please explain.
16. Line 158-188: Results are poorly presented. The authors may utilise tables, charts or graphs to present their results so that they can be easily read and interactive.
17. Discussion: Line 206: the authors claimed the majority of studies with substantial heterogeneity – what analysis did the authors use to address the issue?
18. Line 210-212: Conflicting reports. Please discuss why.
19. Line 213-215: This paragraph seems a little odd. Please elaborate more.
20. In general, the discussion is not well written. The authors should focus on discussing their findings from meta-analysis and how the findings can fill the gaps between the present and previous studies. Also, please rephrase this section, in some parts, it was quite difficult to follow. Use connective words to make it easy to read.
21. Conclusion: the authors may need to elaborate more on how their study will be beneficial to other researchers, clinicians or readers?
22. Please review the use of spacing, punctuation marks and grammar throughout the manuscript. For example, there should be a space between a number and its unit, the use of en dash (–) instead of a hyphen (-) to denote range, the use of a hyphen in some places (compound words, etc).
23. The writing style could be improved. In some place, it was quite difficult to follow, whereas other place was okay.
I consider the manuscript is sufficiently comprehensive and can be considered for publication in the Antibiotics journal after these issues have been properly addressed.
My sincere congratulations to all Authors.
Comments on the Quality of English Language
Please review the use of spacing, punctuation marks and grammar throughout the manuscript. For example, there should be a space between a number and its unit, the use of en dash (–) instead of a hyphen (-) to denote range, the use of a hyphen in some places (compound words, etc).
The writing style could be improved. In some place, it was quite difficult to follow, whereas other place was okay.
Author Response
- Abstract: Please list all the databases used to search related articles in line 20. Also, please confirm the year 2012 is correct? 2012 to 2022 (10 years) for the comprehensive search? Did the authors mean articles published between Jan 2012 and Sept 2022? If so, please rephrase the sentence.
Thank you for your suggestion, the databases have been added and the sentence regarding the study period has been rephrased for clarity
- Line 25: Organism names should be in italics. Please review the manuscript thoroughly.
Thank you for your comment, we have reviewed all mentions of organisms and italicized their names
- Abstract: Results: Results are too general. Please highlight your important results from the meta-analysis in this section.
- Introduction: Line 52: Perhaps a list of the isolated enzymes would benefit the readers.
- Line 53: Please elaborate more on the mechanism of AMR spread via horizontal gene transfer.
- Line 55: Which regions? This will benefit the readers when reading and comparing with your results later.
- Line 56: Multiple antibiotic families such as?
Comments 4-7: thank you for your suggestions, they have been added to the text for clarity
- Methods: Line 77: again, please verify the year that you performed the searches. 10 years is too long. Is this correct? Or are you referring to these dates as the dates for articles published? Otherwise, you may change it to “for articles published between January 1, 2012 and September 30, 2022.
We have again clarified the study and inclusion period.
- Also, is there any specific reason why the recent articles up until September 2024 are not included in the analysis?
Thank you for your question. The reason recent articles up until September 2024 are not included is due to the time required for writing, data analysis, and the publishing process. This naturally resulted in a cutoff for the literature review before these more recent publications became available.
- Figure 1 (PRISMA flow chart) should be moved to the Method section.
- Line 128: Please write the formula in correct format and centered on the page, not in line with text.
The formula has been formatted correctly, thank you for pointing it out.
- Line 134: It should be I2, not I2.
- Line 135: Use an en dash to denote range, not a hyphen. Please review the manuscript thoroughly.
Comments 12 and 13: Thank you for your suggestion, formatting errors have been fixed
- Figure 1: Please list the number of studies identified in each database used.
- Figure 1: studies excluded (n = 653), based on what criteria? Please explain.
Thank you for your comment. The 653 studies were excluded based on the screening of their titles and abstracts. Only studies that advanced to the final stage of selection had specific reasons for exclusion documented.
- Line 158-188: Results are poorly presented. The authors may utilise tables, charts or graphs to present their results so that they can be easily read and interactive.
- Discussion: Line 206: the authors claimed the majority of studies with substantial heterogeneity – what analysis did the authors use to address the issue?
Thank you for your suggestion, further comments on this issue have been added to the limitations section.
- Line 210-212: Conflicting reports. Please discuss why.
Thank you for your comment. The seeming conflict arises because, while Brazil, Argentina, and Colombia are better represented in the data, the overall information across LAC remains limited, especially due to the lack of mandatory reporting in some regions. Even in the more represented countries, data on pediatric CRE-BSI is often incomplete.
- Line 213-215: This paragraph seems a little odd. Please elaborate more.
Thank you for your suggestion, that paragraph has been expanded for clarity.
- In general, the discussion is not well written. The authors should focus on discussing their findings from meta-analysis and how the findings can fill the gaps between the present and previous studies. Also, please rephrase this section, in some parts, it was quite difficult to follow. Use connective words to make it easy to read.
Thank you for your comments. The overall English redaction and flow of the study has been revised.
- Conclusion: the authors may need to elaborate more on how their study will be beneficial to other researchers, clinicians or readers?
Thank you for your suggestion, we expanded the Contributions of this Study section
- Please review the use of spacing, punctuation marks and grammar throughout the manuscript. For example, there should be a space between a number and its unit, the use of en dash (–) instead of a hyphen (-) to denote range, the use of a hyphen in some places (compound words, etc).
Thank you for your suggestion, formatting errors have been fixed
- The writing style could be improved. In some place, it was quite difficult to follow, whereas other place was okay.
Thank you for your suggestion, we have thoroughly reviewed the manuscript to improve redaction.
Reviewer 2 Report
Comments and Suggestions for Authors
The systematic review addressed a critical global health issue, specially focusing on the impact of Carbapenem-resistant Enterobacteriaceae bloodstream infections (CRE-BSI) in the pediatric population within Latin America and the Caribbean (LAC). Author highlights the high-risk factors associated with CRE-BSI, including those with prior antibiotic exposure, immunocompromised states and invasive procedures underline the urgent call for the improved surveillance and stewardship programs. Author also recognized the geographical variability in resistance patterns and infection sources of CRE-BSI, which is important for tailoring interventions to specific local contexts and challenges. This review article could be open the new window for further research to address the gaps in knowledge and improve the understanding of CRE-BSI in pediatric populations, as well as valuable for guiding the future investigations. I appreciate the study undertaken; this systematic review article could be considered for publication after some modifications. Here are my suggestions.
1. Acknowledgment of heterogeneity among studies and the lack of national data due to insufficient report in LAC countries. However, there is limited discussion on how this heterogeneity might affect the overall findings and conclusion. A more detailed analysis of the variability in study methodologies, number of patients, and outcomes could provide a clearer understanding regarding to the CRE-BSI.
2. Author appropriately mentioned that prior use of carbapenem increases the risk of CRE infection in pediatric and adult populations. It would be better to provide more detailed comparisons in antibiotics use between pediatric and adult populations. This will help to understand how the patterns in pediatric population differ from those in the adult populations.
3. Discussion on the risk factors, such as carbapenem use, immunocompromised status, invasive procedures, prior colonization and underlying conditions including metabolic disorders and neutropenia, are well-explained. It would be appreciated to add the specific data or examples from the published studies could provide further understanding on these risk factors and its outcomes.
4. Author revealed that intra-abdominal infections as a predominant source of CRE-BSI, which is remarkable. Why this certain source of infection is more prevalent in hospital settings? Author should include the detailed analysis of this source of infection in discussion.
5. Author noted the resistance mechanisms, such as KPC is predominance followed by NDM and OXA. How the prevalence of these resistance mechanisms effect the treatment options of CRE-BSI and its outcomes on the basis of geographic variability? It would be appreciated if author include these details.
6. As author highlights the antimicrobial stewardship as a key strategy for preventing the emergence and spread of CRE infections. Adding specific examples of successful interventions from other geographical regions could provide the practical guidance for applying this strategy in LAC region.
7. Author discussed the high cost of healthcare system and economic burden are another relevant aspect of CRE infections. It would be better to add the specific cost data or comparison with the cost of other infections could provide an understanding of the financial impact on CRE infections.
Comments on the Quality of English Language
The language of the manuscript can be improved by correcting the grammatical mistakes and typos throughout the manuscript.
Author Response
- Acknowledgment of heterogeneity among studies and the lack of national data due to insufficient report in LAC countries. However, there is limited discussion on how this heterogeneity might affect the overall findings and conclusion. A more detailed analysis of the variability in study methodologies, number of patients, and outcomes could provide a clearer understanding regarding to the CRE-BSI.
Thank you for your suggestion, further comments on this issue have been added to the limitations section.
- Author appropriately mentioned that prior use of carbapenem increases the risk of CRE infection in pediatric and adult populations. It would be better to provide more detailed comparisons in antibiotics use between pediatric and adult populations. This will help to understand how the patterns in pediatric population differ from those in the adult populations.
Thank you for your comment. While we do mention carbapenem resistance and its associated risk factors, we believe that comparing antibiotic use between pediatric and adult populations is not the focus of our study.
- Discussion on the risk factors, such as carbapenem use, immunocompromised status, invasive procedures, prior colonization and underlying conditions including metabolic disorders and neutropenia, are well-explained. It would be appreciated to add the specific data or examples from the published studies could provide further understanding on these risk factors and its outcomes.
- Author revealed that intra-abdominal infections as a predominant source of CRE-BSI, which is remarkable. Why this certain source of infection is more prevalent in hospital settings? Author should include the detailed analysis of this source of infection in discussion.
Thank you for your suggestion. We have added relevant sources on this topic in the discussion.
- Author noted the resistance mechanisms, such as KPC is predominance followed by NDM and OXA. How the prevalence of these resistance mechanisms effect the treatment options of CRE-BSI and its outcomes on the basis of geographic variability? It would be appreciated if author include these details.
Thank you for your comment. While the prevalence of resistance mechanisms like KPC, NDM, and OXA is important, it is not the focus of our current study. While we do have data on patients and resource use, information on resistance mechanisms comes from separate bulletins. We could not find regional data to conduct an analysis by area, as the data sources are distinct and not integrated for this purpose. However, geographic variability does play a role in treatment options and outcomes, and this is certainly a topic for further exploration.
- As author highlights the antimicrobial stewardship as a key strategy for preventing the emergence and spread of CRE infections. Adding specific examples of successful interventions from other geographical regions could provide the practical guidance for applying this strategy in LAC region.
We appreciate your suggestion. While this paper doesn't cover specific examples of successful antimicrobial stewardship interventions, we are working on a second paper that explores this in more detail for the LAC region.
- Author discussed the high cost of healthcare system and economic burden are another relevant aspect of CRE infections. It would be better to add the specific cost data or comparison with the cost of other infections could provide an understanding of the financial impact on CRE infections.
Thank you for your comments. While the cost of CRE infections and their economic system are indeed important aspects to consider, this exceeds the scope of our review, as a cost analysis would be needed to fully analyze this.
Reviewer 3 Report
Comments and Suggestions for Authors
General Comment for Rejection:
The manuscript presents several methodological and conceptual issues that raise concerns about the clarity of the study's objectives, the appropriateness of the selected studies, and the validity of the statistical analyses performed. These factors compromise the overall quality and scientific merit of the paper. Additionally, the manuscript appears to lack up-to-date data, and the interpretation of results is limited by inadequate transparency in the reporting of methodologies and findings. For these reasons, I would recommend rejecting the manuscript at this stage.
Specific Comments:
Line 65-69: The study’s target is vague and covers a broad range. Please clarify the specific epidemiological data you seek, and be precise regarding the type of meta-analysis conducted.
Line 21-27: The abstract should include the results of the meta-analysis to provide readers with a clear summary of the findings.
Line 75-78: The literature search was only conducted up to September 2022. Given that almost two years have passed, the search should be updated to include the most recent studies.
Line 90-95: Please specify which studies were included in more detail, as epidemiological standards require clear categorization of the studies used.
Line 94: What is the rationale behind using a 19-year age cutoff? Please justify the reasoning behind this choice.
Line 97: It appears that a proportional meta-analysis was performed, yet comparative studies were included. Why were comparative studies, typically interventional, used in a context meant to estimate prevalence? This mixture of observational and interventional studies may not provide reliable results for prevalence calculations.
Line 127-130: Please explain why the Freeman-Tukey transformation was selected over other transformations commonly used in meta-analysis.
Line 135-136: Revise the interpretation of the I² statistic to ensure consistency with standard meta-analytic interpretation guidelines.
Results Section: The results are incomplete and unclear. It is concerning that studies with very small sample sizes (1-2 patients) are included in the meta-analysis. This compromises the robustness of the findings.
PROSPERO Registration: In the PROSPERO protocol, studies with more than 10 patients were to be included, but the results section includes studies with fewer patients. Please clarify this discrepancy.
Table 1: The information provided in Table 1 is extremely limited. Consider expanding the details of the included studies, such as the number of participants, outcomes, and follow-up periods.
Pooled Estimates: The manuscript provides several pooled estimates but lacks additional important information, such as forest plots, I² values, and the number of studies used for each outcome. This lack of detail makes the results difficult to interpret and overly exhaustive.
Author Response
Thank you for your comments and suggestions, they have been taken into account to improve our work overall
Round 2
Reviewer 1 Report
Comments and Suggestions for Authors
I consider the manuscript is sufficiently comprehensive and can be considered for publication in the Antibiotics journal after minor editing of English language has been carried out.
My sincere congratulations to all Authors.
Comments on the Quality of English LanguageI consider the manuscript is sufficiently comprehensive and can be considered for publication in the Antibiotics journal after minor editing of English language has been carried out.
Author Response
Thank you for your comments. The last version was improved and edited by a native English editor.
Reviewer 2 Report
Comments and Suggestions for Authors
This review could be acceptable after some modification.
1. Plagiarism of the manuscript should not excess than 20%
2. Author not added the national data as per suggestion. National data is crucial for this study.
2. Author should add the example of successful antimicrobial stewardship interventions in this manuscript, instead of adding in second paper or next paper.
3. Other answers by the author is justified.
Author Response
Thank you for your valuable feedback and suggestions. We have addressed your comments as follows:
1. Regarding the instances of potential plagiarism: We have thoroughly revised the flagged sentences. It's worth noting that many of these passages were derived from previous studies conducted by our research team or individual investigators involved in this project. We have ensured proper attribution and originality in the revised manuscript. In this new version (version 3), we have incorporated all the suggestions from you and other reviewers to avoid plagiarism.
2. Concerning the request for additional national registry data: We appreciate your insightful suggestion. Our study presents data extracted from epidemiological bulletins published by reference laboratories across the region. It's crucial to highlight that these bulletins have significant limitations. Table 3 in our manuscript compiles the available national information from Latin America at the time our systematic review was completed. This underscores a critical gap in our current knowledge base.
Our aim is to draw attention to the urgent need for robust epidemiological surveillance data on CRE bloodstream infections in pediatric populations across our region. While reference laboratories conduct epidemiological surveillance based on samples sent by healthcare centers, the information is not specifically disaggregated for the target population of our systematic review.
3. Regarding the successful implementation of antimicrobial stewardship programs: thank you for your suggestion. We add in the new version local and regional experience about ASP.
Reviewer 3 Report
Comments and Suggestions for Authors
Thank you for your revisions; however, I could not find a point-by-point response to my previous comments. As key concerns remain unaddressed, I regret to say that I must recommend rejection of the manuscript in its current form.
Author Response
We sincerely appreciate your insightful feedback and recommendations. We have carefully considered your input to enhance the quality of our manuscript:
Line 65-69: The study’s target is vague and covers a broad range. Please clarify the specific epidemiological data you seek, and be precise regarding the type of meta-analysis conducted.
Answer:
Thank you for your comment. At the end of the background section we state “This systematic review and meta-analysis aims to comprehensively examine epidemiological data, risk factors, clinical manifestations, microbiological profiles, resistance patterns, and resource utilization associated with CRE-BSI in children from LAC, incorporating both national and regional epidemiological reports.” We believe this statement is clear enough regarding the type of epidemiological outcomes explored.
Moreover, in the Methods sections we state “Proportion meta-analyses were conducted to analyze the data” to clarify which type of meta-analysis were conducted.
Line 21-27: The abstract should include the results of the meta-analysis to provide readers with a clear summary of the findings.
Answer:
Thank you. The results of the meta-analysis were included in the abstract.
Line 75-78: The literature search was only conducted up to September 2022. Given that almost two years have passed, the search should be updated to include the most recent studies.
Answer:
Thank you very much for your comment, which will be the subject of a future study. We needed this time to proccess and analyzed the information to find founds for publication.
Line 90-95: Please specify which studies were included in more detail, as epidemiological standards require clear categorization of the studies used.
Answer:
Thank you for your comments. We add in elegibility criteria all types of studies included.
Line 94: What is the rationale behind using a 19-year age cutoff? Please justify the reasoning behind this choice.
Answer:
Thank you for your comment. For most countries in the Region, hospitalizations of pediatric patients
correspond to the age group of 0 to 18 years.
Line 97: It appears that a proportional meta-analysis was performed, yet comparative studies were included. Why were comparative studies, typically interventional, used in a context meant to estimate prevalence? This mixture of observational and interventional studies may not provide reliable results for prevalence calculations.
Answer:
Thank you for your comment. The included studies were observational, not interventional.
Line 127-130: Please explain why the Freeman-Tukey transformation was selected over other transformations commonly used in meta-analysis.
Answer:
- Proportion meta-analyses are frequently used in epidemiology to estimate the burden of disease. They are usually based on transformed proportions using Freeman‐Tukey double arcsine transformations (FTT). Doi SA, Xu C. The Freeman-Tukey double arcsine transformation for the meta-analysis of proportions: Recent criticisms were seriously misleading. J Evid Based Med. 2021;14: 259–261.
- In a recent Abstract presented in the 2023 Cochrane Colloquium (see ref below) our group conducted GLMM and FTT over a large dataset of proportions from a living systematic review and meta-analysis about safety, immunogenicity, and effectiveness of COVID-19 vaccines for pregnant people (https://safeinpregnancy.org/lsr/) applying recommended safeguards (using corrected statistical packages: Metan in Stata) and other approaches (GLMM and Metaprop in R):
We used the following safeguards:
a) avoiding the use of the average of the double arcsine and its variance for synthesis;
b) using the inverse of the variance of the pooled FTT proportion
c) modifying the confidence intervals to prevent numerical inaccuracies.
- Afterward, we compared results for MAs with few/several studies, and for Vaccine outcomes/Adverse effects
- We concluded that the FTT is the most reliable approach and remains the preferred transformation in proportion meta-analysis
Ciapponi A , Bardach A , Glujovsky D , Castellana N Is the Freeman‐Tukey double arcsine transformation a reliable approach? for proportion meta-analysis <Abstracts accepted for the 27th Cochrane Colloquium, London, UK. Cochrane Database of Systematic Reviews 2023; (1 Supp 1):[abstract number]. https://doi.org/10.1002/14651858.CD202301>)
Line 135-136: Revise the interpretation of the I² statistic to ensure consistency with standard meta-analytic interpretation guidelines.
Answer:
Thank you. However in the Methods section we interpret this statistic as follows: “We calculated the I2 statistic to measure the proportion of the overall variation attributable to between-study heterogeneity. An I2 >60–70% was considered substantial heterogeneity, and below 30% was a low level of heterogeneity” This is in line with standard MA guidelines.
Results Section: The results are incomplete and unclear. It is concerning that studies with very small sample sizes (1-2 patients) are included in the meta-analysis. This compromises the robustness of the findings.
Answer:
Thank you. We consider that these are some of the limitations of the study that were registered. The
analysis was carried out with the best available evidence for the Region.
PROSPERO Registration: In the PROSPERO protocol, studies with more than 10 patients were to be included, but the results section includes studies with fewer patients. Please clarify this discrepancy.
Answer:
Thank you. We agree with the comment, but as a working group we decided to carry out the analysis
with the best available evidence and include all studies regardless of the number of patients reported
due to the low availability of data in the Region.
Table 1: The information provided in Table 1 is extremely limited. Consider expanding the details of the included studies, such as the number of participants, outcomes, and follow-up periods.
Answer:
Thank you very much for your comment. The information corresponding to each study is detailed in the different sections of the manuscript.
Pooled Estimates: The manuscript provides several pooled estimates but lacks additional important information, such as forest plots, I² values, and the number of studies used for each outcome. This lack of detail makes the results difficult to interpret and overly exhaustive.
Answer:
Thank you very much for your comment, we describe those that we consider relevant to the study.
Round 3
Reviewer 3 Report
Comments and Suggestions for Authors
Thank you for addressing the reviewer’s comments and providing detailed responses. We appreciate the effort you have put into refining the manuscript. However, after carefully reviewing your revisions, there remain several critical areas where the responses do not fully resolve the concerns raised. As a result, I propose that the manuscript be considered for rejection unless these issues can be comprehensively addressed in a future revision:
1. Outdated Literature Search: The literature search is almost two years old, and the rationale provided for not updating it is insufficient. Given the fast-evolving nature of scientific research, an outdated search significantly reduces the relevance and accuracy of your findings.
2. PROSPERO Protocol Adherence: The inclusion of studies with fewer than 10 patients, which deviates from your stated PROSPERO protocol, is concerning. Although you mention the limited availability of data in the region, this discrepancy requires a more robust justification.
3. Table 1 Details: While you mention that the information is provided in other sections of the manuscript, the request was to expand Table 1 itself to include more specific details such as the number of participants, outcomes, and follow-up periods.
4. Inclusion of Studies with Very Small Sample Sizes: Including studies with only 1-2 patients compromises the robustness of the meta-analysis. While you acknowledge this as a limitation, the justification provided is not sufficient.
Author Response
Outdated Literature Search: The literature search is almost two years old, and the rationale provided for not updating it is insufficient. Given the fast-evolving nature of scientific research, an outdated search significantly reduces the relevance and accuracy of your findings.
Answer: we updated the literatury research up to September 30th 2024
PROSPERO Protocol Adherence: The inclusion of studies with fewer than 10 patients, which deviates from your stated PROSPERO protocol, is concerning. Although you mention the limited availability of data in the region, this discrepancy requires a more robust justification.
Answer: due to scarce of available data, we have decided to include all the papers which mention our data of interest. Including case series with <10 patients, was the only way to cover comprehensively this information. This protocol deviation was necesary to include the available data from our region.
Table 1 Details: While you mention that the information is provided in other sections of the manuscript, the request was to expand Table 1 itself to include more specific details such as the number of participants, outcomes, and follow-up periods.
Answer: thank you for your suggestion. We added data related to sex, age, underlying diseases, and mortality in the table 1 according to your suggestion.
4. Inclusion of Studies with Very Small Sample Sizes: Including studies with only 1-2 patients compromises the robustness of the meta-analysis. While you acknowledge this as a limitation, the justification provided is not sufficient.
Answer: thank you for your comments. We decided include these studies for final analysis and this acknowledge was recognized in the discussion.
We submtit a new version with updated information.